# A Deep-Learning-Based Secure Routing Protocol to Avoid Blackhole Attacks in VANETs [note 1]

**DOI:** 10.3390/s23198224

**Published:** 2023-10-02

**Authors:** Amalia Amalia, Yushintia Pramitarini, Ridho Hendra Yoga Perdana, Kyusung Shim, Beongku An

**Affiliations:** 1Department of Software and Communications Engineering in Graduate School, Hongik University, Sejong City 30016, Republic of Korea; amalia@mail.hongik.ac.kr (A.A.); yushintia@mail.hongik.ac.kr (Y.P.); hendra@mail.hongik.ac.kr (R.H.Y.P.); 2School of Computer Engineering and Applied Mathematics, Hankyong National University, Anseong City 17579, Republic of Korea; kyusung.shim@hknu.ac.kr; 3Department of Software and Communications Engineering, Hongik University, Sejong City 30016, Republic of Korea

**Keywords:** deep learning, secure routing, clustering, blackhole, vehicular ad-hoc networks

## Abstract

Vehicle ad hoc networks (VANETs) are a vital part of intelligent transportation systems (ITS), offering a variety of advantages from reduced traffic to increased road safety. Despite their benefits, VANETs remain vulnerable to various security threats, including severe blackhole attacks. In this paper, we propose a deep-learning-based secure routing (DLSR) protocol using a deep-learning-based clustering (DLC) protocol to establish a secure route against blackhole attacks. The main features and contributions of this paper are as follows. First, the DLSR protocol utilizes deep learning (DL) at each node to choose secure routing or normal routing while establishing secure routes. Additionally, we can identify the behavior of malicious nodes to determine the best possible next hop based on its fitness function value. Second, the DLC protocol is considered an underlying structure to enhance connectivity between nodes and reduce control overhead. Third, we design a deep neural network (DNN) model to optimize the fitness function in both DLSR and DLC protocols. The DLSR protocol considers parameters such as remaining energy, distance, and hop count, while the DLC protocol considers cosine similarity, cosine distance, and the node’s remaining energy. Finally, from the performance results, we evaluate the performance of the proposed routing and clustering protocol in the viewpoints of packet delivery ratio, routing delay, control overhead, packet loss ratio, and number of packet losses. Additionally, we also exploit the impact of the mobility model such as reference point group mobility (RPGM) and random waypoint (RWP) on the network metrics.

## 1. Introduction

Vehicular ad-hoc networks (VANETs) have emerged as an important component in the intelligent transportation system (ITS) field [1,2]. In a VANET, vehicles are equipped with devices that enable wireless communication, effectively turning each vehicle into a mobile node within the network. This allows them to communicate with each other (vehicle-to-vehicle or V2V communication) and with infrastructure such as traffic signals or roadside units (vehicle-to-infrastructure or V2I communication) [3,4]. Routing in VANETs pertains to the process of finding and maintaining efficient routes for data transmission between these mobile nodes. Due to the high mobility and dynamic nature of the network, it can lead to frequent route disconnection and network topology changing [5]. Therefore, the routing protocol is essential for efficient data transmission between the source node and the destination node.

In VANETs, the clustering protocol is considered as one of the possible approaches to enhance network stability and node connectivity by forming similar node characteristics [6,7,8]. In detail, the cluster consists of three types of nodes: the cluster head (CH), cluster member (CM), and gateway (GW) [9,10]. The CH works as the central node in a cluster and is responsible for managing communication among its CMs. It gathers data from CMs and forwards it to other CHs or the GW. CMs are regular nodes within the cluster, associated with a CH, and follow the CH’s decisions to efficiently use network resources and save energy. The GW is a special node that connects different clusters or extends the communication range beyond a single cluster. It acts as a bridge forwarding data between clusters when CHs need to communicate or when a CH is out of range from other CHs, ensuring that messages reach their intended destinations.

Security is a critical issue in VANETs due to the potential for serious accidents. A blackhole attack that disrupts the data transmission process by dropping all the packets instead of forwarding them is especially serious [11,12]. When other nodes try to communicate with the destination, the blackhole node sends fake route replies (fake-RREP) by having the freshest and shortest or most efficient route towards the destination node [13,14]. However, instead of forwarding the packets to the actual destination, the blackhole node intentionally drops or discards the packets. As a result, the data packets are lost, and the source nodes never receive any response from the destination node. This interruption in data transmission can lead to a significant disruption in communication within the network and causes an increase in the end-to-end delay and low packet delivery ratio (PDR).

To address the problem above, we use a deep neural network (DNN) model for detecting blackhole attacks in VANETs. Deep learning (DL) is a powerful machine learning technique that can analyze complex data patterns through DNN [15]. This approach offers advantages over conventional methods, especially in identifying the distinguishing features between normal and malicious network behavior. DL efficiently processes high-dimensional data leading to more accurate detection [16]. According to the authors in [17], we can actively avoid blackhole nodes by using DL to select the optimal route based on fitness function values and improve the optimization speed, thereby reducing delays and enhancing network performance. Additionally, since DL is one of the compact mapping functions, the trained DL can find the relation between input data and output data without additional processes. Thus, it can be adapted to detect whether the next node is a blackhole attack node or not. Usually, an iteration-based algorithm is used to optimize fitness values but it spends so much time finding the optimal value than the DNN model. Therefore, by using the DNN model, we can reduce delays in the optimization process and ultimately improve the performance of the overall system.

### 1.1. Related Works and Motivation

Routing in VANETs is an active area of research due to the dynamic and distributed nature of these networks. To manage the challenges of routing in networks on a large scale, clustering has emerged as a commonly used approach. Clustering plays a crucial role in ensuring high stability and reducing control overhead within a network. One of the first and most referenced protocols in this area is LEACH [18], which used a randomized rotation of nodes as CH to evenly distribute the energy load among the sensors in the network. However, the cluster is randomly selected from all sensor nodes, regardless of their energy levels, affecting network longevity. Deterministic algorithms such as the highest-degree algorithm [19] or the lowest-ID algorithm [20] use static node properties such as node ID or degree (number of neighbors) to select CH. These are simpler but may not adapt well to network changes. In addition, the authors in [19] do not consider other important metrics such as energy levels, which can impact the network’s overall performance, while the authors in [20] do not take into account any metric other than the ID itself. The authors in [21] introduce a particle swarm optimization (PSO)-based clustering algorithm for VANETs. This algorithm not only establishes the rules for clustering but also designs a routing algorithm within the cluster and among different clusters to improve routing efficiency in V2V communication. More recent studies have proposed the use of artificial intelligence techniques for CH selection. For example, some of the research leverages fuzzy logic [22] and genetic algorithms [23] to optimize cluster head selection. However, the iteration-based algorithm, as mentioned above spends so much time finding the optimal cost or fitness function. Hence, using a machine learning technique such as DL can provide a more efficient and accurate solution compared to iteration-based algorithms, especially when dealing with large-scale, high-dimensional, and complex problems.

In addition to the clustering process, security stands as another important aspect of routing, ensuring the safe transmission of data in VANETs. One of the most common attacks that has a negative impact on network performance is the blackhole attack. A variety of methods were suggested to prevent and identify blackhole attacks. The authors in [24] used a modification to the AODV protocol to prevent blackhole attacks in mobile ad hoc networks (MANETs). This protocol used an intrusion detection system to detect the blackhole attacks. However, the false-positive detection of blackhole nodes in a harsh environment may occur. The authors in [25] proposed the detection and prevention of blackhole attacks in VANETs. The authors made the algorithm to scan all the existing routes for the chance blackhole attacks exist. If the route with a blackhole node is found, then the route is ignored and another route is searched for. This algorithm exhibits the capability to identify and mitigate the effects of black hole attacks within highly mobile VANETs. Despite this, the processing time required by this algorithm is significant, leading to an increased end-to-end delay. The authors in [12] proposed SVODR to avoid blackhole attacks in VANETs. An encrypted random number field is introduced into the RREQ packet, which is then broadcast to all adjacent nodes. When the source node receives the RREP, it validates the destination sequence number in its own routing table against the destination sequence number and the encrypted or decrypted random numbers in the RREP. If the RREP’s destination sequence number exceeds the destination sequence number in the source vehicle’s routing table and the random numbers from both operations match, the node is considered a legitimate node. If not, the vehicle is flagged as malicious. A potential limitation of this approach is that it necessitates additional fields in control packets for cryptography algorithms, which require more resources. This can lead to substantial routing overhead and increased end-to-end delay.

The authors in [26] modeled SSAE and softmax classifier deep network schemes for DDoS attack detection. The key features were extracted from a dataset and their dimensionality was reduced using the SSAE model. These selected features were then fed into a softmax classifier to detect attacks. Although this method significantly decreases training time, it does not enhance security when it is applied to hardware with limited resources. The authors in [27] proposed SVM for IDS. SVM is a well-known machine learning algorithm that is widely used for pattern classification problems. This method prevented the modification of classified events, but it did not include the large amounts of data obtained via vehicular communication to improve the SVM’s performance. In [28], to improve security, the authors proposed the CMEHA-DNN deep learning-based intrusion detection model for the purpose of detecting Sybil attacks in VANETs. The Sybil attack is successfully identified by this method. These works [26,27,28] did not focus on blackhole attacks in VANETs. However, from the above works, we can conclude that machine learning techniques, especially DL, can be used to avoid attacks in VANETs.

Table 1 shows the full comparison of the existing approach and our proposed approach to easily compare and identify the research gap of our study. It is evident that the existing schemes have numerous limitations. Thus, we propose a deep learning approach that is utilized in various aspects of networking, including clustering, routing, and blackhole detection which has not been explored by other authors in the previous studies. This leads to the enhancement of both the security and efficiency of network routing. By employing DL, we are able to leverage the inherent complexities of the data and patterns presented within the network, leading to more accurate results compared to conventional methods. The DL approach also can reduce delays in the optimization process and ultimately improve the performance of the overall system.

Based on the above, three fundamental questions arise, which will be addressed in this article: (1) In VANETs, how do we protect the confidential packet from a blackhole attack? (2) Can the data-driven method be utilized instead of the iteration-based method? (3) Is it possible to manage mobile nodes to improve route stability?

### 1.2. Contributions and Organization

To address the security vulnerability posed by blackhole attacks in VANETs, we propose a deep-learning-based secure routing (DLSR) protocol to avoid blackhole attacks. In addition, we also propose a deep-learning-based clustering (DLC) protocol to enhance route connectivity and reduce control overhead. One of the key features of our proposed routing protocol is its ability to adaptively choose the flexibility between secure routing and normal routing based on network security conditions. This flexibility allows the protocol to maintain a balance between security and routing performance, ensuring that the network remains both safe from blackhole attacks and efficient in terms of performance. The main contribution of this paper can be summarized as follows:We propose the DLSR protocol to avoid blackhole attacks in VANETs. The proposed DLSR protocol utilizes DL on each node to select secure routing or normal routing depending on network security conditions to maintain a balance between security and routing performance. More specifically, DL is used to identify any suspicious behavior of a node and to determine the best possible next hop based on its fitness function value.We propose the DLC protocol as the underlying structure to enhance route connectivity and reduce control overhead in VANETs. We consider node distance, node speed, node direction, and remaining energy to form a cluster.We design the DNN model to optimize the fitness function in both the routing and clustering process. By using DNN, the proposed DLSR protocol can optimize the weights of each parameter, such as remaining energy, distance, and hop count, which leads to choosing the best route against blackhole attacks; while in the DLC protocol, DNN is used to optimize the weights of each parameter, such as cosine similarity, cosine distance, and remaining energy, which leads to electing CH.The simulation results show that the proposed DLSR with DLC protocol can establish a route that is much more resistant to blackhole attacks than the benchmark protocol. Moreover, as the clustering organizes nodes into groups, the proposed DLC protocol enables stronger connectivity and reduces control overhead. Additionally, in order to demonstrate the efficiency of the proposed protocol, we also compare it with different mobility models, namely, the reference point group mobility (RPGM) and random waypoint (RWP).

The rest of the paper is organized as follows: Section 2 provides a brief background of blackhole attacks in VANETs. Section 3 explains the proposed system model that consists of the basic concept of the proposed routing and clustering protocol, the proposed DLC protocol, and the proposed DLSR protocol. Section 4 explains the proposed DL framework. Section 5 discusses the performance evaluation that consists of simulation environments and network parameters, performance metrics, and numerical results, and Section 6 concludes the paper.

## 2. Overview of Blackhole Attack

One kind of network security risks that affects wireless networks, particularly VANETs, is the blackhole attack. In the blackhole attack, a malicious node can attract all packets by falsely claiming a fresh route to the destination and then absorbing them (dropping all packets) without forwarding them to the destination [2,29]. This causes other nodes in the network to route their packets through the malicious node, which then drops or consumes the packets instead of forwarding them as expected. It is also known as a sequence number attack because it uses sequence numbers for dropping the packet [30]. The sequence number is a numbering scheme kept by the source node of the route request (RREQ) and route reply (RREP) to help maintain the freshness of the routing information.

In the routing process, the source node starts a route discovery if the route toward the destination is not found or if the route is no longer active. In this case, the source node broadcasts RREQ to the neighbor nodes (NN). If the destination node receives this RREQ, it responds to the source node with RREP containing updated information such as source identifier, destination identifier, sequence number, broadcast ID, and hop count. A blackhole node can exploit this process to present itself as having the freshest route to the destination by quickly responding with a fake RREP with a significantly higher sequence number than the normal nodes in the network without checking its routing table. A higher value of destination sequence number means a fresher route [31]. In this way, a blackhole node becomes part of the route. Then, the source selects this route, and the blackhole node starts to drop the packets, degrading the overall performance of the VANETs. Dropping these packets in highly dynamic VANETs could result in road fatalities, accidents, traffic jams, and congestion. Hence, our research focuses on addressing the blackhole attack in VANETs and proposes an efficient solution to avoid it.

A blackhole attack can be detected by studying the behavior of blackhole nodes in a network. Therefore, by using the behavioral characteristics of a blackhole attack, we can make a detection system to classify the type of node between a blackhole node and a normal node [11,32,33,34], which can be summarized as follows:The blackhole node has a higher destination sequence number.The blackhole node has a lower number of hop counts toward the destination node.The blackhole node has higher remaining energy.The blackhole node responds to all RREQ packets by sending an RREP packet.The blackhole node never broadcasts any RREQ packet received.The blackhole node drops the received data packet in the network.

## 3. System Model

In this section, we introduce the network topology used for VANETs in urban areas, incorporating clustering to enhance route connectivity. The network nodes are organized into clusters, grouping nodes with similar characteristics. Each cluster is led by a CH, which is responsible for managing communications within the cluster. A key concern in this topology is the threat of blackhole attacks among the vehicle nodes. These attacks falsely claim to have the optimal route to the destination and then drop packets without forwarding them, causing a disruption in the network performance. To address this challenge, we propose a DLSR protocol, which can predict and counter blackhole attacks by establishing secure routes to the destination. Additionally, we employ the DLC protocol, which groups vehicle nodes into clusters based on distance, speed, direction, and remaining energy. This approach ensures strong connectivity and enhances the overall performance of the network.

### 3.1. The Basic Concept of The Proposed Secure Routing and Clustering Protocol

In this subsection, we present the basic concept of our proposed routing protocol. The proposed routing protocol can be divided into two processes, called clustering by using the DLC protocol and routing by using the DLSR protocol, that can avoid a blackhole attack. These processes can be summarized as follows:**Clustering Process:** In the proposed DLC protocol, network nodes are grouped into clusters based on shared information such as distance, speed, direction, and remaining energy. CH is selected based on the highest remaining energy. DL is used to optimize the weights of the fitness function for selecting the most optimal CH so that a node can join a cluster as CM effectively and transmit its data efficiently. The DNN model takes input parameters such as cosine similarity, cosine distance, and remaining energy to calculate the optimized weight of the fitness function. GW is also considered to improve route connectivity and reduce control overhead in the clustering process.**Routing Process:** Following the clustering process, a source node broadcasts an RREQ packet to find the destination node. However, a blackhole node may send a fake RREP to deceive the source node into believing it has the optimal route. To overcome this, our proposed DLSR protocol uses DL to detect and select the optimal route for avoiding blackhole attacks. When a node receives RREP packets, it feeds the relevant feature data into the DNN model. The model then processes this information and generates an output, classifying the received data as originating from a blackhole node or a normal node. Then, based on the fitness value, the routing protocol can dynamically choose to operate in either secure or normal routing mode. Here, we also optimize the weight of the fitness function by using DL. Figure 1 shows an illustration when an intermediate node receives two RREP packets from the NN. One of the RREP packets received is a fake RREP from a blackhole node. In this case, the proposed DLSR protocol employs secure routing by detecting it through DL and calculating the fitness function. Therefore, the proposed DLSR protocol can establish the secure route from *S*–$CH_{1}$–$CH_{2}$–$CH_{3}$–$CH_{m}$–*D* to avoid a blackhole attack.

### 3.2. The Proposed DLC Protocol: The Underlying Structure

The proposed clustering protocol is called the DLC protocol as an underlying structure, as shown in Figure 2. The proposed DLC protocol uses a DNN model to optimize the fitness function weights for selecting a CH to follow by a CM. Here, we consider three factors to form a cluster, namely, cosine similarity, cosine distance, and remaining energy. Figure 3 shows the flowchart of the proposed DLC protocol. The procedure for forming a cluster and electing the CH is as follows:**Step 0: Initialization**

Each node activates and operates independently once the simulation begins.


**Step 1: The Dissemination of Node Information**


Each node $n_{k}$ periodically estimates its remaining energy information. Then, node $n_{k}$ generates and broadcasts information (INFO) packets to its NN periodically to advertise its node information with NN. The following fields are included in the INFO packet:$$\langle \mathrm{Type},S_{\mathrm{ID}},D_{\mathrm{ID}},E\rangle$$
where Type represents packet type, $S_{\mathrm{ID}}$ represents source node ID, $D_{\mathrm{ID}}$ represents destination node ID, and E represents the remaining energy of the node, respectively. Then, go to **Step 2**.


**Step 2: Cluster Heads Candidate Selection**


The CH is chosen by the candidate node from all NN with the highest remaining energy. This can be mathematically expressed as [35]
(1)$$k^{*}=\arg\max_{k\in {\mathrm{NN}}_{k}^{*}\cup \left\{k\right\}}E_{k},$$
where $E_{k}$ indicates the remaining energy of the $k^{th}$ node. The node with the highest remaining energy is chosen as a CH because all communication must pass through the CH. Therefore, it provides robust connectivity between CH and cluster members (CM). If *k* = $k^{*}$, the node $n_{k}$ becomes CH, then go to **Step 3**. Otherwise, go to **Step 4**.


**Step 3: The Dissemination of Cluster Head Information**


When $n_{k}$ becomes the CH, $n_{k}$ generates and broadcasts the cluster head information (CHI) packet to advertise it to its NN. The following fields are included in the CHI packet:$$\langle \mathrm{Type},S_{\mathrm{ID}},D_{\mathrm{ID}},\mathrm{Pos},S,\mathrm{Dir}\rangle$$
where Type represents the packet type, $S_{\mathrm{ID}}$ represents the source node ID, $D_{\mathrm{ID}}$ represents the destination node ID, Pos represents the node position $\left(x_{k},y_{k}\right)$, S represents the node speed, and Dir represents the node direction, respectively. Then, go to **Step 4**.


**Step 4: Determination of Gateway Nodes**


The GW is selected when $n_{k}$ is located in between more than one CH. In this case, the CHs from the neighbor may send $n_{k}$ several CHI packets. Therefore, $n_{k}$ becomes the GW and goes to **Step 5.1**. Otherwise, $n_{k}$ becomes a CM node and goes to **Step 5.2**.



**Step 5: Determination of Cluster Members**
The procedure for the node $n_{k}$ receiving either one or multiple CHI packets is as follows:
-**Step 5.1:** If node $n_{k}$ receives multiple CHI packets, it selects the CH among the CH candidates to follow based on the maximum fitness function value. Input parameters such as cosine similarity, cosine distance, and remaining energy are taken into account. The proposed fitness function can be expressed as (2a)$$\max_{\left\{x_{1},x_{2},x_{3}\right\}}F_{k}=x_{1}{\mathrm{CS}}_{k}+x_{2}{\mathrm{CD}}_{k}+x_{3}E_{k},$$
(2b)$$s.t\;x_{1}+x_{2}+x_{3}=1,$$
(2c)$${\mathrm{CS}}_{k}\geq {\mathrm{CS}}_{th},$$
(2d)$${\mathrm{CD}}_{k}\leq {\mathrm{CD}}_{th},$$
(2e)$$E_{k}\geq E_{th},$$ where ${\mathrm{CS}}_{k},{\mathrm{CD}}_{k}$, and $E_{k}$ are the cosine similarity, cosine distance, and remaining energy, respectively. Equation ([Disp-formula FD2b-sensors-23-08224]) indicates the total weight that must be equal to one. Furthermore, ([Disp-formula FD2c-sensors-23-08224]) explains that the cosine similarity must be greater than equal to the cosine similarity threshold, ([Disp-formula FD2d-sensors-23-08224]) explains that the cosine distance must be lower than equal to the cosine distance threshold, and ([Disp-formula FD2e-sensors-23-08224]) explains that the remaining energy of the CH must be greater than equal to energy threshold, respectively. To find the optimal weight in ([Disp-formula FD2b-sensors-23-08224]), we use the proposed DNN model, which is elaborated on in detail in the forthcoming Section 4.1. Using a DNN model to optimize the weights of a fitness function can provide a more efficient and accurate solution compared to iteration-based algorithms, especially when dealing with large-scale, high-dimensional, and complex problems. Moreover, the iteration-based algorithm spends so much time finding the optimal weights compared to the DNN model. This efficiency can lead to reduced delays in the optimization process and ultimately improve the performance of the overall system. Here, we consider cosine similarity, which serves as a measure of similarity between the movement patterns of vehicles. Vehicles with high cosine similarity have similar movement patterns and, therefore, can be grouped together in the same cluster. The cosine similarity between two nodes can be defined as [36]
(3)$$\mathrm{CS}\left(k,l\right)=\frac{\sum_{l=1}^{N}{\vec{V}}_{k}{\vec{V}}_{l}}{\sqrt{\sum_{k=1}^{N}{\vec{V}}_{k}^{2}}\sqrt{\sum_{l=1,l\neq k}^{N}{\vec{V}}_{l}^{2}}},$$
where ${\vec{V}}_{k}$ and ${\vec{V}}_{l}$ are the $k^{th}$ and $l^{th}$ node’s vector information list, respectively. Each node ${\vec{V}}_{k}$ is associated with vector information metric values such as speed, direction, and location. It can be defined as ${\vec{V}}_{k}=\left({\vec{V}}_{1},{\vec{V}}_{2},...,{\vec{V}}_{l}\right)$, where ${\vec{V}}_{k}$ indicates a value that denotes the association between nodes. Furthermore, we consider the cosine distance between mode $n_{k}$ and its neighbor. We may control CM to create a more stable CM from the perspective of mobility by taking into account the highest cosine similarity under the constrained communication distance. Therefore, the cosine distance can be defined as [37]
(4)CD(k,l)={1−CS(k,l)}.
Here, we also consider the remaining energy of the node. By considering the remaining energy, we can design energy-efficient clustering protocols that take into account the energy consumption of nodes, ultimately prolonging their lifetime. To establish an accurate and reliable energy model, we reference the work of [7,8], which provide comprehensive insights into energy-efficient approaches in wireless networks. Then, the selected CM can be mathematically formulated as
(5)$$l^{*}=\arg\max_{l}\left\{F_{k},F_{l}\right\},$$
where $F_{k}$ indicates the fitness function value of node $n_{k}$, and $F_{l}$ indicates the fitness function value of neighbor CH near node $n_{k}$ and goes to **Step 5.2**.-**Step 5.2:** If $n_{k}$ chooses the CH, then the join cluster (JC) packet is sent by $n_{k}$ to the CH. This happens when node $n_{k}$ receives only one CHI packet; it directly sends the JC packet. The following fields are included in the JC packet:$$\langle \mathrm{Type},S_{\mathrm{ID}},D_{\mathrm{ID}},\mathrm{Stat}\rangle$$ where Type represents the packet type, $S_{\mathrm{ID}}$ represents the source node ID, $D_{\mathrm{ID}}$ represents the destination node ID, and Stat represents the node status, whether it will be GW or not, respectively. Then, go to **Step 6**.




**Step 6: Cluster Members and Cluster Heads Table Updates**



Node $n_{k}$ sends the JC packet to CH. When the CH receives the JC packet, it replies with an accept cluster (AC) packet. The following fields are included in the AC packet:$$\langle \mathrm{Type},S_{\mathrm{ID}},D_{\mathrm{ID}}\rangle$$
where Type represents the packet type, $S_{\mathrm{ID}}$ represents the source node ID, and $D_{\mathrm{ID}}$ represents the destination node ID, respectively. Furthermore, node $n_{k}$ updates the CM table, the CH table is also updated, and then the cluster is formed. The obtained clustering table can be summarized in Table 2, where $N_{\mathrm{ID}}$ is the node ID, Stat is the node status (CM, GW, or CH), ${\mathrm{CH}}_{\mathrm{ID}}$ is the CH ID to which the node belongs, and CM is a list of the node IDs of CM belonging to the same cluster as the current node, respectively.

Taking into account the dynamic and random movement of nodes in VANETs, it is common for nodes to frequently switch over between clusters. Nevertheless, with the proposed clustering algorithm, member nodes including the source and GW nodes exhibit minimal switching between clusters. This stability can be associated with the similarity in mobility patterns among CM. Table 3 shows the summary of the packet list involved in the proposed DLC protocol.

### 3.3. The Proposed DLSR Protocol

The proposed routing protocol is called the DLSR protocol. We assume that the clustering process has already been completed and is running periodically during the routing process. Figure 4 shows an illustration when a blackhole node appears in the network. Each node employs DL to detect the presence of a blackhole node. If a blackhole node is identified along the route, then the fitness value will have a higher value than the other route, and secure routing is employed to avoid it. The flowchart of the proposed DLSR protocol is shown in Figure 5, which can be divided by the route request process and the route reply process.

#### 3.3.1. Route Request Process

In this sub-subsection, we explain in detail the RREQ process of the proposed DLSR protocol as follows:**Step 1: Initialization**

If the route between a source node *S* and a destination node *D* does not exist, the source node *S* starts the route establishment process.


**Step 2: Source Generates and Sends RREQ Packet**


The source node *S* (a CM) wants to send a message to a destination node *D* located in another cluster. In this case, *S* generates the RREQ packet with a unique ID and a sequence number and then sends it to the ${\mathrm{CH}}_{i}$ in the cluster if *S* does not have the routing information to the destination. The RREQ packet contains the following fields:$$\langle \mathrm{Type},S_{\mathrm{ID}},D_{\mathrm{ID}},S_{\mathrm{seq}},D_{\mathrm{seq}},B_{\mathrm{ID}},H\rangle$$
where Type represents the packet type, $S_{\mathrm{ID}}$ represents the source node ID, $D_{\mathrm{ID}}$ represents the destination node ID, $S_{\mathrm{seq}}$ represents the source sequence, and $D_{\mathrm{seq}}$ represents the destination sequence, which is the number of attempts to confirm control messages. $B_{\mathrm{ID}}$ represents the broadcast ID, which is the number of generating RREQs in the same session at the source, and H is a hop count, respectively. Then, go to **Step 3**.


**Step 3: Intermediate Nodes Operation at CH for RREQ**
When ${\mathrm{CH}}_{i}$ receives the RREQ packet, it checks the freshness of the received RREQ packet and then checks its routing table for a valid route towards *D*. The operation to check the freshness of the packet can be summarized as follows:-**Step 3.1:** If $S_{\mathrm{seq}}$ at the received RREQ is larger than $S_{\mathrm{seq}}$ at the routing table, then go to **Step 3.3**. Otherwise, go to **Step 3.2**.-**Step 3.2:** If $S_{\mathrm{seq}}$ at the received RREQ is equal to $S_{\mathrm{seq}}$ at the routing table, then check again whether the $B_{\mathrm{ID}}$ at the received RREQ is equal to $B_{\mathrm{ID}}$ at the routing table or if $B_{\mathrm{ID}}$ at the received RREQ is larger than $B_{\mathrm{ID}}$ at the routing table and H at the received RREQ plus one is less than H at the routing table or not, then go to **Step 3.3**. Otherwise, the packet will be dropped.-**Step 3.3:** If the node ID in the ${\mathrm{CH}}_{i}$ table is the same as the destination ID in the RREQ packet, then go to **Step 3.4**. Otherwise, go to **Step 3.5**.-**Step 3.4:** ${\mathrm{CH}}_{i}$ records the sender’s ID, updates the routing table, and sends the RREQ packet to the destination using unicast, then goes to **Step 5**.-**Step 3.5:** ${\mathrm{CH}}_{i}$ records the sender’s ID, updates the routing table, and broadcasts RREQ to NN, then goes to **Step 4**.


**Step 4: Intermediate Nodes Operation at GW for RREQ**


When ${\mathrm{GW}}_{i}$ receives the RREQ packet from ${\mathrm{CH}}_{i}$, it records the sender’s ID, updates the routing table, and broadcasts RREQ to ${\mathrm{NN}}_{i}$ by increasing the number of hop counts. **Steps 1–4** are repeated until the RREQ packet reaches the destination node’s CH and goes to the **Route Reply Process**.

#### 3.3.2. Route Reply Process

Following the route request process, in this sub-subsection, we explain in detail the RREP process of the proposed DLSR protocol, as follows:**Step 5: Destination Generates and Sends RREP Packet**

Once the destination node receives the RREQ packet, it records the sender’s ID, updates the routing table, and an RREP packet with an updated destination sequence number is generated. It unicasts the RREP packet to the previous node using the reverse path. The RREP packet contains the following fields:$$\langle \mathrm{Type},D_{\mathrm{ID}},S_{\mathrm{ID}},D_{\mathrm{seq}},E,\mathrm{Pos},H,F\rangle$$
where Type represents the packet type, $D_{\mathrm{ID}}$ represents the destination node ID, $S_{\mathrm{ID}}$ represents the source node ID, $D_{\mathrm{seq}}$ represents the destination sequence, E is the remaining energy, Pos represents the node position $\left(x_{i},y_{i}\right)$, H is a hop count, and F is a fitness function, respectively. Then, go to **Step 6**.


**Step 6: Intermediate Node Operation at GW for RREP**


${\mathrm{GW}}_{i}$ records the sender’s ID and updates the routing table when it receives the RREP packet. Then, ${\mathrm{GW}}_{i}$ forwards the RREP packet to the previous node by using unicast and increasing the number of hop counts, then goes to **Step 7**. Otherwise, it waits until it receives the RREP packet.


**Step 7: Intermediate Node Operation at CH for RREP**
When ${\mathrm{CH}}_{i}$ receives the RREP packets, it checks whether the route information received is from a blackhole node or not. The process can be summarized as follows:-**Step 7.1:** If ${\mathrm{CH}}_{i}$ receives the RREP packet more than once, then go to **Step 7.2**. Otherwise, go to **Step 7.6**.-**Step 7.2:** Classify the RREP packet received by using the DNN model that is elaborated in detail in the forthcoming Section 4.2. This model is used to identify any suspicious behavior of a node. If no blackhole node is detected, go to **Step 7.3**. Otherwise, go to **Step 7.4**.-**Step 7.3:** If the packet is not from a blackhole node, calculate the fitness function for normal routing (without penalty for blackhole nodes). The fitness function can be expressed as
(6a)$$\min_{\left\{x_{1},x_{2},x_{3}\right\}}F_{i}=x_{1}E_{i}+x_{2}{\mathrm{Dist}}_{i}+x_{3}H_{i},$$
(6b)$$s.t\;x_{1}+x_{2}+x_{3}=1,$$
(6c)$$E_{i}\geq E_{\mathrm{th}},$$
(6d)$${\mathrm{Dist}}_{i}\leq D_{\mathrm{th}},$$
(6e)$$H_{i}\leq H_{\mathrm{th}},$$
where $E_{i},{\mathrm{Dist}}_{i}$, and $H_{i}$ are the remaining energy, distance, and hop count, respectively. Equation ([Disp-formula FD6b-sensors-23-08224]) indicates the total weight that must be equal to one. Equation ([Disp-formula FD6c-sensors-23-08224]) explains that the remaining energy of the node must be greater than equal to the remaining energy threshold, ([Disp-formula FD6d-sensors-23-08224]) explains that the distance must be lower than equal to the distance threshold, ([Disp-formula FD6e-sensors-23-08224]) explains that the hop count must be lower than equal to the hop count threshold, respectively. To find the optimal weight in ([Disp-formula FD6b-sensors-23-08224]), we use the proposed DNN model that is elaborated in detail in the forthcoming Section 4.2. Furthermore, to establish an accurate and reliable energy model, we reference the work of [7,8], which provides comprehensive insights into energy-efficient approaches in wireless networks. In addition, we consider the distance between two nodes, which is based on the node position and can be expressed as
(7)$${\mathrm{Dist}}_{\left(i,i^{\prime }\right)}=\sqrt{{\left(x_{i}-x_{i^{\prime }}\right)}^{2}+{\left(y_{i}-y_{i^{\prime }}\right)}^{2}},$$
where $x_{i}$ and $y_{i}$ represents the node location longitude and latitude, respectively. Then, go to **Step 7.5**.-**Step 7.4:** If the packet is from a blackhole node, then we employ secure routing. Here, we introduce the penalty term, as follows:
(8)$$P_{i}=\hat{P}\times \mathrm{blackhole},$$
where $\hat{P}$ is a large positive constant, and blackhole is a binary variable (1 if a blackhole node is detected, 0 otherwise). Thus, the updated fitness function with a penalty term can be expressed as
(9)$$F_{P_{i}}=F_{i}+P_{i},$$
where $F_{i}$ is the original fitness function, and $P_{i}$ is the penalty term. The idea is to make the fitness value higher than a normal node, effectively discouraging the selection of routes with blackhole nodes, then go to **Step 7.5**.-**Step 7.5:** The best route is selected by choosing the minimum fitness function value for data transmission that can be mathematically formulated as
(10)$$i^{*}=\arg\min_{i\in R}F_{i},$$
where $F_{i}$ indicates the fitness function value of the $i^{th}$ route. Then, go to **Step 7.6**.-**Step 7.6:** ${\mathrm{CH}}_{i}$ records the sender’s ID, updates the routing table, and sends the packet to *S* by using unicast and increasing the number of hop counts, then goes to **Step 7.7**.-**Step 7.7:** If *S* receives RREP, then it goes to **Step 8**. Otherwise, wait until *S* receives the RREP packet.


**Step 8: Data Transmission Process: Source Received RREP and Preparing for Data Transmission**


The data transmission is sent by *S* to *D* based on the routing table, which is determined in **Step 1** to **Step 7**.

The routing table can be summarized in Table 4, where PN is a previous node, NX is the next node, F is the fitness function, $N_{\mathrm{ID}}$ is the node ID, $S_{\mathrm{ID}}$ is the source ID, $D_{\mathrm{ID}}$ is the destination ID, $B_{\mathrm{ID}}$ is the broadcast ID, $S_{\mathrm{seq}}$ is the source sequence, $D_{\mathrm{seq}}$ is the destination sequence, and H is the hop count, respectively. Therefore, based on the explanation above, the summary of the packet list involved in the proposed DLSR protocol can be seen in Table 5.

## 4. The Proposed Deep Learning Framework

In this section, we present the proposed DL framework based on the DNN model to obtain the optimal solution, as shown in Figure 6. More specifically, the proposed DL framework is used for solutions to these problems in this paper, as follows:***Problem I***: Finding optimal fitness function value for clustering.***Problem II***: Detecting whether the next node is a blackhole node or not.***Problem III***: Finding the optimal fitness function value for routing.

We consider the general DL framework with two phases: phase 1 is for training the DNN model, and phase 2 is for testing the DNN model, as shown in Figure 6. Figure 6a shows the DNN learning from the generated dataset in the training phase, where the DNN model learns the dataset generated from the conventional method. The conventional method can guarantee convergence at the global optimum. The optimal solutions obtained from the conventional method serve as the target parameters of the DNN model. The DNN will be trained to learn the relationship between input trainable parameters and target parameters. In the training phase, an error will be obtained from comparing the DNN output with the target based on the optimal solution of the conventional method. Then, this error will be minimized by updating the weight and bias on the neurons by using backpropagation, which continues until the iteration is satisfied [38]. After training, the trained DNN model can predict the optimal value with new input data variables, as shown in Figure 6b. By using these processes, the DNN learns to predict the optimal value. However, it can be noted that each DL framework to solve each problem is totally different, which can be explained in the following subsection.

### 4.1. Deep Learning for Clustering

To solve problem I, we propose the architecture of the DNN model to obtain the optimal solution for clustering that is presented in Figure 7. As we can see, the main goal of this approach is to demonstrate the speed, distance, vehicle direction, and the energy remaining to determine the weight value ($x_{1}$, $x_{2}$, $x_{3}$). This is very different from the conventional optimization method for solving the problem (2), which requires some iteration. We use the speed (S), distance (Dist), the vehicle from 0${}^{\circ }$ direction (${\mathrm{Dir}}_{1}$), the vehicle from 180${}^{\circ }$ direction (${\mathrm{Dir}}_{2}$), and the energy remaining (E) with size 1×5 as input parameters for the neural network to learn the complex functions, several hidden layers with numerous hidden neurons connected with symmetric weight, while the output layer is the prediction of $x_{1}$, $x_{2}$, and $x_{3}$. Then, we generate a dataset to train a DNN model for multiple-output regression problems. Table 6 presents the structure of the layers employed in the DNN model, which enhances the system’s performance.

### 4.2. Deep Learning for Routing

In the routing process, we proposed the architecture of the DNN model that is employed twice for the purposes of blackhole node detection and finding the optimal weight that minimizes the routing fitness function value, as detailed below:**Deep learning for blackhole detection:** To solve problem II, we utilize a DNN model for classifying blackhole nodes and normal nodes in VANETs. This makes it possible to choose the flexibility between secure routing and normal routing strategies based on the types of nodes that were identified. Our primary objective is to enhance the security and efficiency of routing in VANETs by detecting and avoiding blackhole attacks. Based on the behavioral characteristics of the blackhole attack mentioned in Section 2, we can define the input parameter to train the DNN model. The input parameters are node identifier ($N_{\mathrm{ID}}$), destination sequence number ($D_{\mathrm{seq}}$), hop counts (H), remaining energy (E), and the node’s position (Pos). The DNN model we employ is a feed-forward neural network with 1×5 dimensional input layers, several hidden layers with numerous hidden neurons connected with symmetric weight, and 1×2 dimensional output layers to classify nodes in real-time as either blackhole nodes ($B_{n}$) or normal nodes ($N_{n}$), which can be seen in Figure 8. Table 7 presents the structure of the layers employed in the DNN model.**Deep learning for finding optimal fitness function values for routing:** To solve problem III, we also utilize a DNN model to find the optimal weight that minimizes the routing fitness function values the same way as in clustering but with different parameters of the input layers. Three factors such as the remaining energy (E), distance (Dist), and hop count (H) of the node are considered for input parameters. To determine the ideal weight values for $x_{1}$, $x_{2}$, and $x_{3}$, we employ a feed-forward neural network with 1×3 dimensional input layers, several hidden layers with numerous hidden neurons connected with symmetric weight, and 1×3 dimensional output layers, as shown in Figure 9. Table 8 presents the structure of the layers employed in the DNN model, which enhances the system’s performance.

## 5. Performance Evaluation

### 5.1. Simulation Environments and Network Parameters

In the conducted simulation, we attempt to gain a deeper understanding of the proposed routing and clustering protocol. In this comparison, we look at how the DLSR protocol performed against the ad hoc on-demand distance vector (AODV) protocols. Table 9 shows the simulation environments and parameters.

We simulate the proposed routing protocol using network simulator 3 (NS3), where the simulations are run for 1000 s with 5 s for each session. We specifically deploy 30, 50, and 100 nodes moving throughout 1000 $m^{2}$× 1000 $m^{2}$ area in an urban scenario. Additionally, we compare two different mobility models, namely, the RPGM model [39,40] and the RWP model [41], to evaluate their performance in VANETs. The RPGM model represents a scenario where vehicles move cohesively in groups, following similar trajectories and maintaining relative proximity to each other. On the other hand, the RWP model represents a scenario where vehicles move randomly and independently, each following its own trajectory without any coordination with other vehicles. The purpose of this comparison is to understand how the choice of mobility model impacts various performance metrics. The mobile nodes’ initial positions are randomly distributed along the street, and they move at different speeds (20, 40, 60, and 80 km/h). Each mobile node has an omnidirectional antenna, and the node’s maximum transmission range is set to 250 m (approximately). The medium access control (MAC) layer is modeled based on the IEEE 802.11 standard. A received signal strength indicator (RSSI) threshold of −80 dBm is used to define the communication range more practically, ensuring realistic communication scenarios in the simulation. Furthermore, to evaluate the performance of the DL framework, we calculate the accuracy of the predicted optimal value against the output data from the test set. This is accomplished by computing the root mean square error (RMSE) in our proposed DL framework. The RMSE can be expressed as [35]
(11)$$\mathrm{RMSE}=\sqrt{\frac{1}{n}\sum_{k=1}^{n}{\left(\delta_{\left(k\right)}-{\tilde{\delta }}_{\left(k\right)}\right)}^{2}},$$
where *n* represents the number of samples in the test set, $\delta_{\left(k\right)}$ represents the the predicted optimal value of the *k*-th observation in the dataset, and ${\tilde{\delta }}_{\left(k\right)}$ represents the actual value for the *k*-th observation in the dataset, respectively. A smaller RMSE indicates a closer match between the predictions and the observations.

### 5.2. Performance Matrices

The performances of the proposed clustering protocols (DLC) and routing (DLSR) are evaluated in terms of the following metrics [8]:PDR refers to the ratio of the number of the received data packet at a destination node over the number of the transmitted packet at a source node.The routing delay refers to the average time to establish the route between a source node to a destination node per session.The control overhead refers to the average number of control packets to establish a route per session per node.The average number of cluster head changes refers to the average number of cluster head changes per cluster per session.Packet loss ratio refers to the ratio of the number of packet losses to the total number of sent packets.

These specified metrics collectively provide a comprehensive evaluation of the proposed DLC and DLSR performance. They address essential aspects of VANETs operation, including communication quality, efficiency, and network stability. By analyzing the protocol’s performance across these metrics, researchers can assess its suitability for real-world VANETs scenarios and make informed decisions about its deployment and optimization.

### 5.3. Numerical Results

In this subsection, we present the simulation results of the proposed DLC and DLSR protocol. First, we exploit the impact of the number of iterations to search for the best fitness value. Then, in our experiment, we create 100.000 datasets, utilizing 90% for training and the remaining 10% for validation. Furthermore, we produce 100 distinct datasets to evaluate the performance of our trained DNN model. The objective is to examine the ability of the DNN model to predict the optimal value provided with an unfamiliar dataset. Table 10 shows the parameters we considered to examine DNN training.

Figure 10a illustrates the convergence of the fitness function value for clustering, reaching an optimal value within the 8th iteration using the conventional algorithm. Moreover, as the vehicle speed increases, the fitness value also increases, indicating consistent convergence within the 8th iteration despite higher speeds. On the other hand, Figure 10b displays the convergence of the fitness function value for routing, reaching an optimal value within the 10th iteration. Furthermore, with the increasing vehicle speed, the fitness value decreases consistently within the 10th iteration. This proves the ability of the algorithm to handle an increase in speed without affecting its convergence time.

Figure 11 shows the RMSE of the DL framework with different numbers of hidden layers and hidden neurons. In Figure 11a, we analyze the impact of the number of hidden neurons on the DL framework with the different numbers of hidden layers for clustering. As we can see in Figure 11a, the value of RMSE with one hidden layer decreases from 0.5295 to 0.1817; with two hidden layers, it decreases from 0.4756 to 0.0917; and with four hidden layers, it decreases from 0.3220 to 0.08 when the number of hidden neurons increases from 2 to 250, respectively. Figure 11b shows the impact of the number of hidden neurons on the DL framework with the different numbers of hidden layers for blackhole detection. As can be seen in Figure 11b, the value of RMSE with one hidden layer decreases from 0.01124 to 0.00098; with two hidden layers, it decreases from 0.00855 to 0.00074; and with three hidden layers, it decreases from 0.00554 to 0.00039 when the number of hidden neurons increases from 2 to 100, respectively. In Figure 11c, we analyze the impact of the number of hidden neurons on the DL framework with the different numbers of hidden layers for routing. Figure 11c reveals that the RMSE value with one hidden layer decreases from 0.1887 to 0.0361; with two hidden layers, it decreases from 0.1764 to 0.0259; and with three hidden layers, it decreases from 0.1673 to 0.0101 when the number of hidden neurons increases from 2 to 200, respectively. Therefore, from the results shown in Figure 11, we can conclude that the performance of a DNN model improves with an increase in the number of neurons it contains. In addition, a larger number of hidden layers in the DNN model also contributes to better performance, as opposed to models with fewer hidden layers.

Figure 12 provides a comparison of the execution times between the conventional methods and the DL method, considering varying quantities of data. The DL method achieves impressively short execution times, even when the quantity of data increases. In contrast, the conventional method requires a significantly longer execution time under the same conditions. As we can see in Figure 12, the conventional method requires 18 s to find the optimal solution dealing with 50 data, while the DL method accomplishes the same task in just one second for the clustering scheme. In the routing scheme, the conventional method requires 11 s to find the optimal solution dealing with 50 data, while the DL method accomplishes the same task in just one second. The reason is that DNNs are designed to efficiently handle large amounts of data. The use of multiple layers with neurons allows for complex modeling and efficient computation. Furthermore, once the DNN model is trained, the prediction phase is very fast, which reduces execution time.

Figure 13 shows the comparison of the average number of cluster head changes per session as a function of node speed, serving as an evaluation of cluster stability. From Figure 13, it is evident that as the node speed increases, the average number of cluster head changes also increases. This can be attributed to the frequent changes in node location due to the higher speeds, resulting in disruptions to the clustering process. Furthermore, as the number of nodes in the network increases, the average number of cluster head changes also increases. This can be attributed to the higher network density, which leads to more frequent changes in cluster heads. In addition, we are evaluating the effect of the mobility model (RPGM and RWP) on the average number of cluster head changes within the network. As we can see in Figure 13, the RPGM model exhibits a lower average number of cluster head changes compared to the RWP model. The reason is that in the RPGM model, vehicles move cohesively in groups, which means they tend to follow similar trajectories and move together as a unit. On the other hand, in the RWP model, vehicles move randomly and independently, each following its own trajectory without any coordination with other vehicles. Thus, the RPGM model performs more stable and cohesive clusters, resulting in a lower average number of cluster head changes compared to the RWP model, where vehicle movement is independent. However, it is important to note that the average number of cluster head changes remains less than one. This indicates that, on average, there is less than one cluster head change in each session. Hence, the proposed clustering algorithm demonstrates a high level of stability as the number of cluster head changes is kept minimal.

The comparison of the PDR as a function of node speed is shown in Figure 14. The PDR decreases as the node speed increases. The possible reason is that the network becomes more unstable since the node is more dynamically moved when the node speed increases, leading to the occurrence of packet loss. However, we can see that the DLSR with DLC (DLSR+DLC) protocol has a markedly higher level of PDR than other protocols. As a result, the DLSR with DLC protocol turned out to be the most reliable in terms of PDR. In addition, the DLSR+DLC protocol with the RPGM model shows a high PDR compared to the RWP model. The reason is that RPGM provides a more controlled and structured mobility pattern, leading to better communication and more stable network connectivity compared to RWP.

Figure 15 compares the routing delay, which includes the consumed time for cluster construction per session as a function of node speed. As we can see in Figure 15, when node speed increases, the delay increases as well. The reason is that when the speed increases, frequent network topology changes lead to frequent interruptions in data transmission. Again, the protocol functioning under the RPGM model proves to be more stable than that under the RWP model. The reason is that the coordinated group movement in RPGM minimizes abrupt shifts in the network topology, consequently reducing the delay in establishing routes. Conversely, CH and GW are the only nodes involved in the proposed DLSR+DLC protocol, which leads to enhancing the efficiency of the routing process. Thus, the DLSR+DLC protocol with the RPGM model can transmit packets with the least amount of delay compared to other protocols.

The control overhead, which includes the control overhead for cluster construction, is compared as a function of node speed in Figure 16. As we can see in Figure 16, the control overhead slightly increases as the speed increases. The reason is that when the number of node speeds increases, nodes can be broken more easily, leading to an increase in the control overhead. Furthermore, when comparing the two mobility models, it is clear that the RPGM protocol shows lower control overhead due to high stability than the RWP model. In contrast to other scenarios, the control overhead in our proposed routing protocol can be decreased by the DLC protocol. The decrease in control overhead is significantly lower in the case of the DLSR+DLC protocol than in other scenarios, as the DLC protocol only involves CH and GW nodes in the routing operation. Thus, it was shown that the DLSR+DLC protocol with the RPGM model may increase connectivity while simultaneously being the most stable in terms of control overhead.

The focus of our attention is now on security issues. In order to evaluate the impact of node speed on the network performance, we conduct a comparison analysis of the packet loss ratio and number of packet losses under various scenarios, as shown in Figure 17. Figure 17a displays the average packet loss ratio, which is calculated as the ratio of the number of loss packets to the total number of sent packets. It is evident that when the node speed increases, the average packet loss ratio also increases. Similarly, in Figure 17b, when the node speed increases, the average number of packet losses also increases. The observed behavior can be attributed to the increased node speed, causing the node’s location to change more frequently. Consequently, packets are more likely to be sent directly to the blackhole node, resulting in a higher packet loss rate. On the other hand, by using the RPGM model over the RWP model, we can handle movement-related issues better. The RPGM model ensures the predictable and collective movement of nodes. Thus, the RPGM model leads to fewer route changes, more stable connections, and ultimately, a lower chance of packet loss. However, clustering plays a significant role in reducing the number of links between nodes. Therefore, the proposed DLSR+DLC protocol under the RPGM model showcases enhanced network security, making it a promising and effective approach from a network security perspective.

Finally, we analyze how the number of nodes affects the network performance metric, as shown in Figure 18. In Figure 18a, we analyze the PDR as the function of node speed with various numbers of nodes in the proposed DLSR with DLC protocol. The PDR slightly decreases as the node speed increases. However, the PDR slightly increases as the number of nodes increases. This behavior can be attributed to the more predictable movement patterns in the RPGM model, leading to less frequent route changes and fewer packet losses. Furthermore, Figure 18b shows the routing delay as the function of node speed with various numbers of nodes in the proposed DLSR with DLC protocol. As the node speed and the number of nodes increase, a slight increase in routing delay can be observed, although not significantly. This can be attributed to the fact that with more nodes and higher speeds, the number of hops in the routing path increases, leading to a slightly longer routing process. Figure 18c shows the control overhead as a function of node speed with various numbers of nodes in the proposed DLSR with DLC protocol. It demonstrates that an increase in both node speed and the number of nodes leads to higher control overhead but not significantly. This observation can be explained by two possible reasons. First, with a higher number of nodes, the network density increases, causing a higher frequency of packet transmissions, thus, contributing to the increased control overhead. Second, the increase in node speed results in more frequent node mobility, which can lead to more frequent link breakages and route rediscoveries, further contributing to the increase in control overhead. Moreover, to address the control overhead issue, we consider a clustering protocol. The DLC protocol creates clusters with similar mobility patterns using cosine similarity and cosine distance metrics, effectively reducing the occurrence of frequent cluster head changes, especially at higher speeds. Despite these challenges, it is evident that the proposed DLSR+DLC protocol, paired with the RPGM mobility model, exhibits impressive scalability. It effectively enhances the PDR, reduces routing delay, and mitigates control overhead as the number of nodes increases.

From these results, the combination of the proposed methods (DLSR + DLC) can establish a secure route under various mobility models against the blackhole attack. In detail, the DL framework is properly utilized to solve these problems that maximize the fitness function value for clustering and blackhole node detecting and minimize the fitness function value for routing, respectively. The proposed clustering algorithm can enhance the route connectivity, and the proposed DLSR routing protocol can avoid the blackhole attack without additional detecting steps, as well as support network scalability.

## 6. Conclusions

In this paper, we proposed a DLSR and DLC protocol in VANETs to establish a secure route against blackhole attacks. The proposed DLSR protocol utilized DL at each node to select secure routing or normal routing depending on network security conditions by training a DNN model to detect abnormal node behavior. This enabled nodes to select the most suitable next-hop based on their fitness function value, which is also achieved through DNN by optimizing the weights of each parameter such as remaining energy, distance, and hop count. Furthermore, to enhance route connectivity and reduce control overhead we proposed a DLC protocol that employed DNN to optimize the weights of each parameter such as cosine similarity, cosine distance, and remaining energy, which led to electing a cluster head candidate. Additionally, we compared the impact of the mobility model, i.e., the RPGM and RWP model on network metrics. The numerical results showed that the proposed DLSR+DLC protocol with the RPGM model can establish a secure route that improved node connectivity against blackhole attacks. Overall, the proposed DLSR+DLC protocol with the RPGM model efficiently established the cluster and route, which led to improvements in network metrics such as PDR, routing delay, control overhead, packet loss ratio, and number of packet losses.

## Figures and Tables

**Figure 1 sensors-23-08224-f001:**
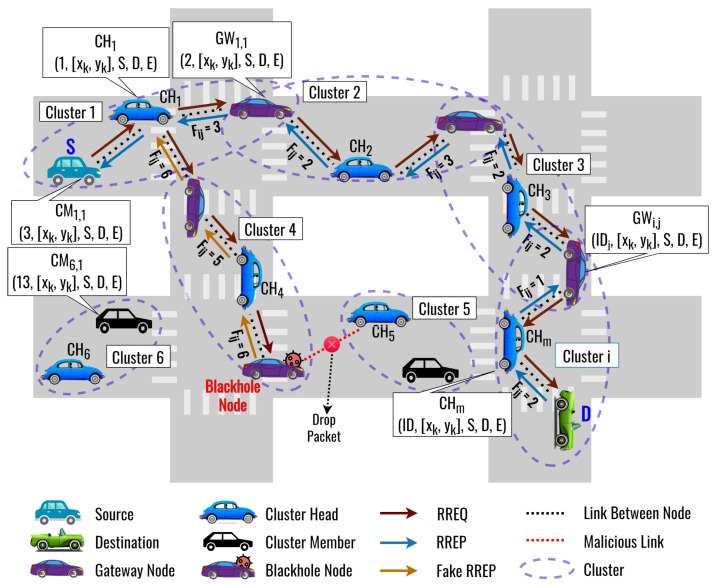
The basic concepts of the proposed DLSR protocol.

**Figure 2 sensors-23-08224-f002:**
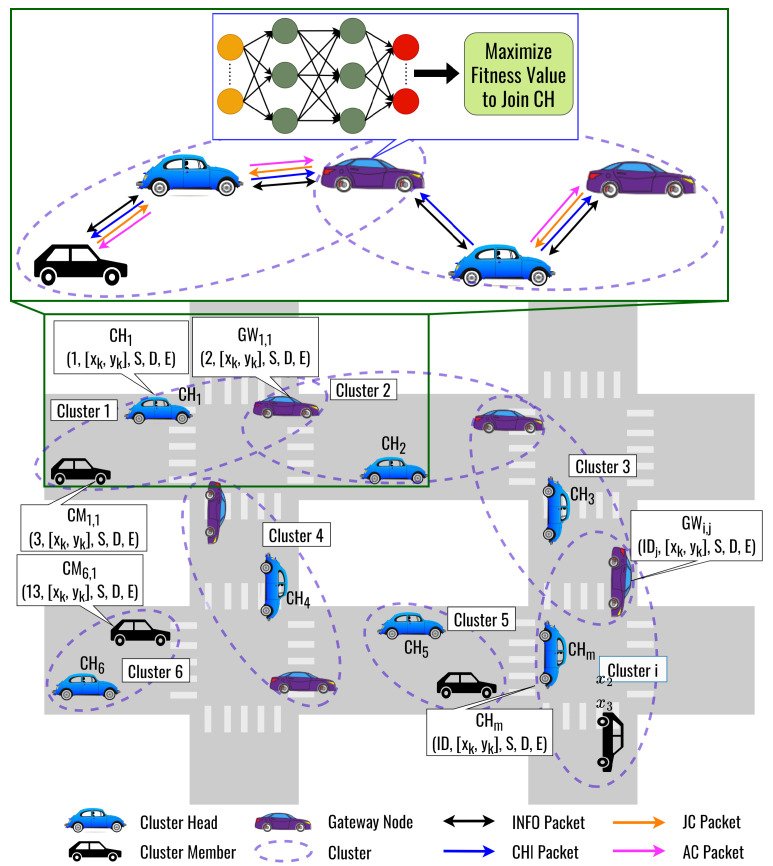
The Basic Concept of the DLC Protocol.

**Figure 3 sensors-23-08224-f003:**
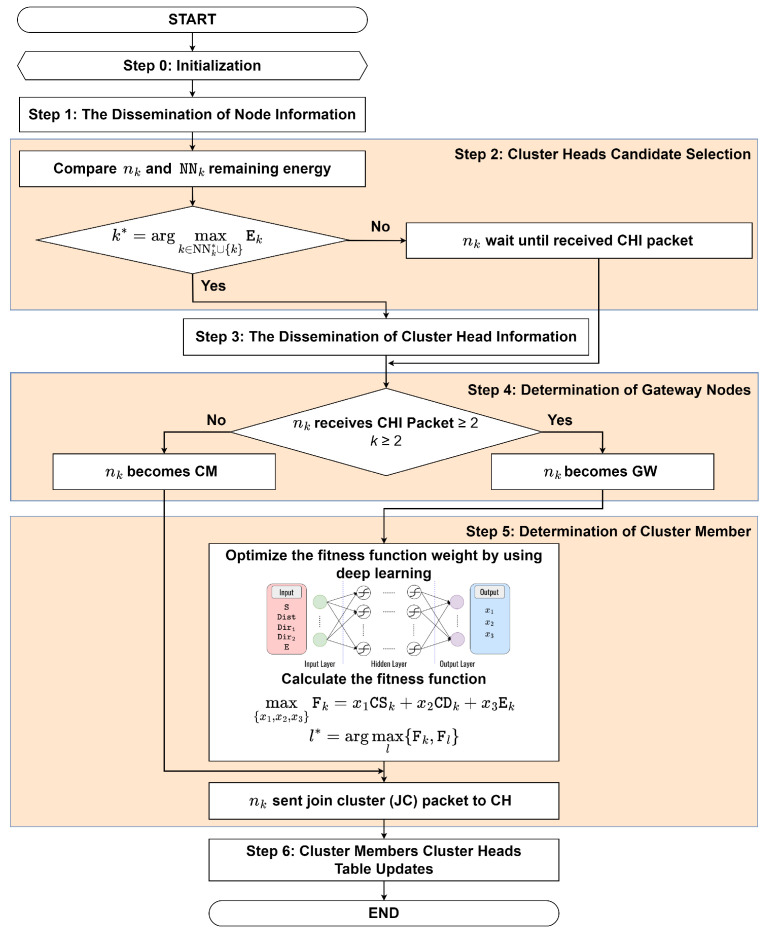
The Flowchart of the Proposed DLC Protocol.

**Figure 4 sensors-23-08224-f004:**
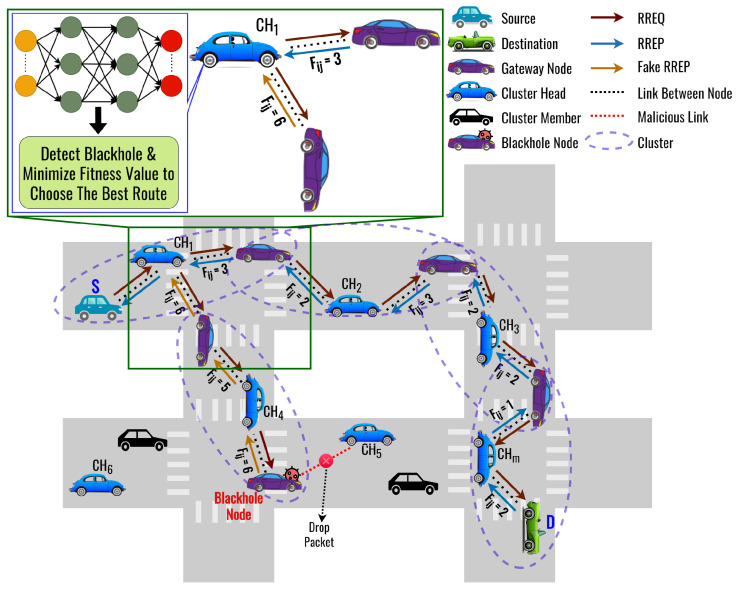
The Basic Concept of the DLSR protocol.

**Figure 5 sensors-23-08224-f005:**
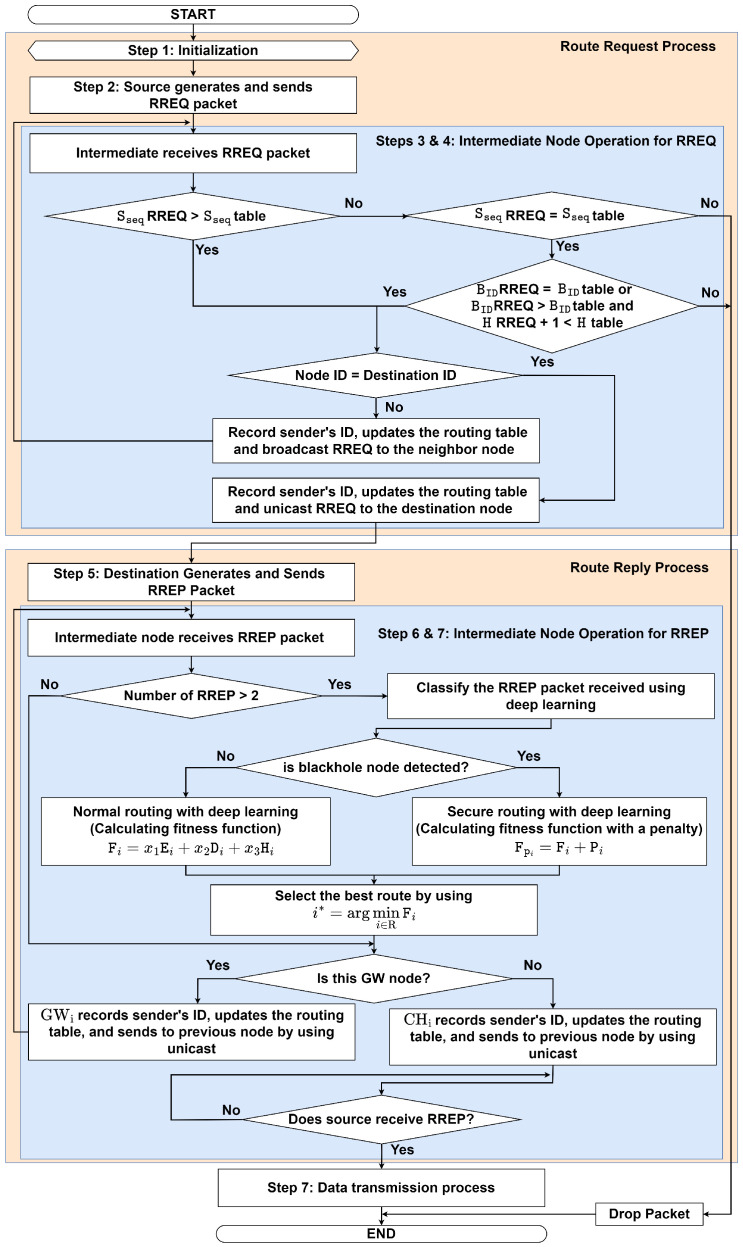
The Flowchart of the Proposed DLSR Protocol.

**Figure 6 sensors-23-08224-f006:**
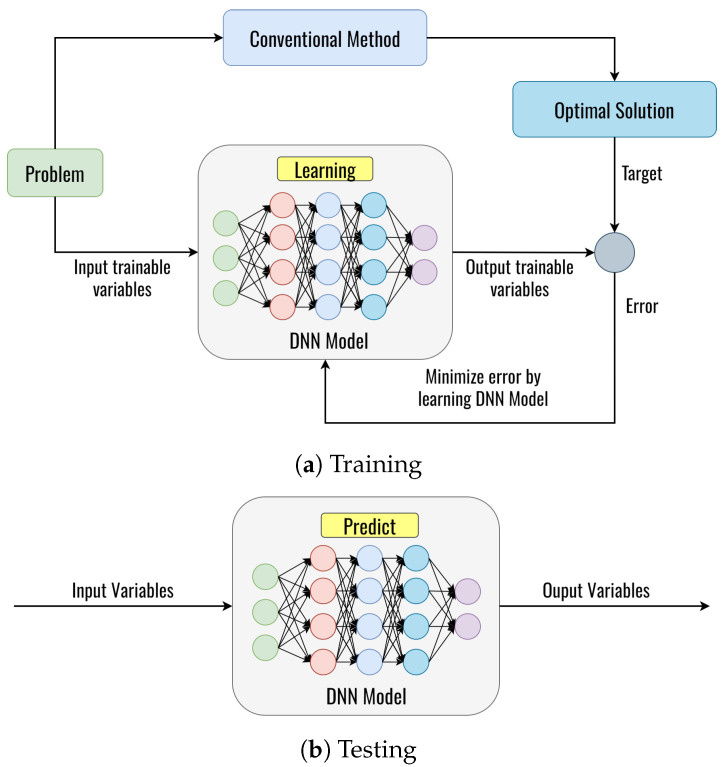
The DL framework.

**Figure 7 sensors-23-08224-f007:**
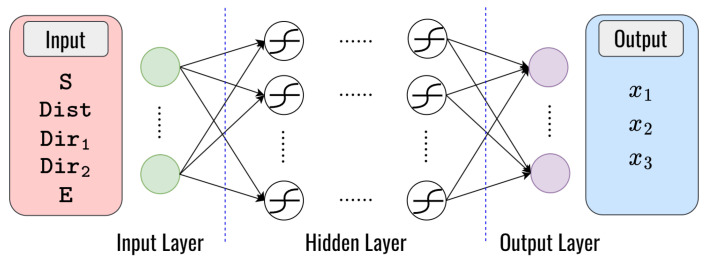
The structure of the DNN model for clustering.

**Figure 8 sensors-23-08224-f008:**
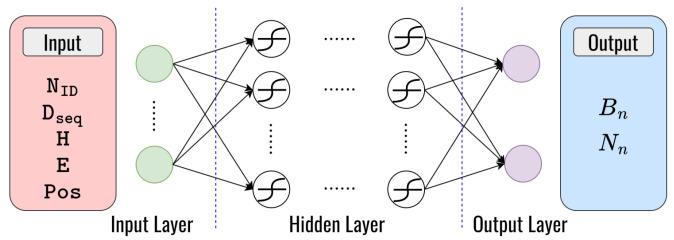
The structure of the DNN model for node classification.

**Figure 9 sensors-23-08224-f009:**
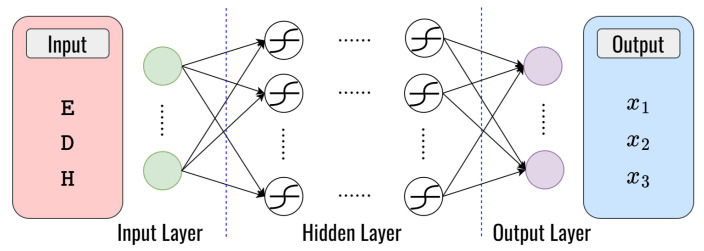
The structure of the DNN model for routing.

**Figure 10 sensors-23-08224-f010:**
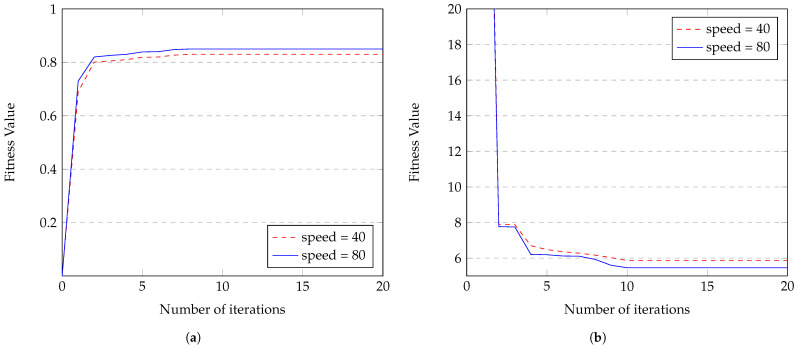
Convergence using the conventional algorithm for (**a**) clustering fitness value maximization and (**b**) routing fitness value minimization.

**Figure 11 sensors-23-08224-f011:**
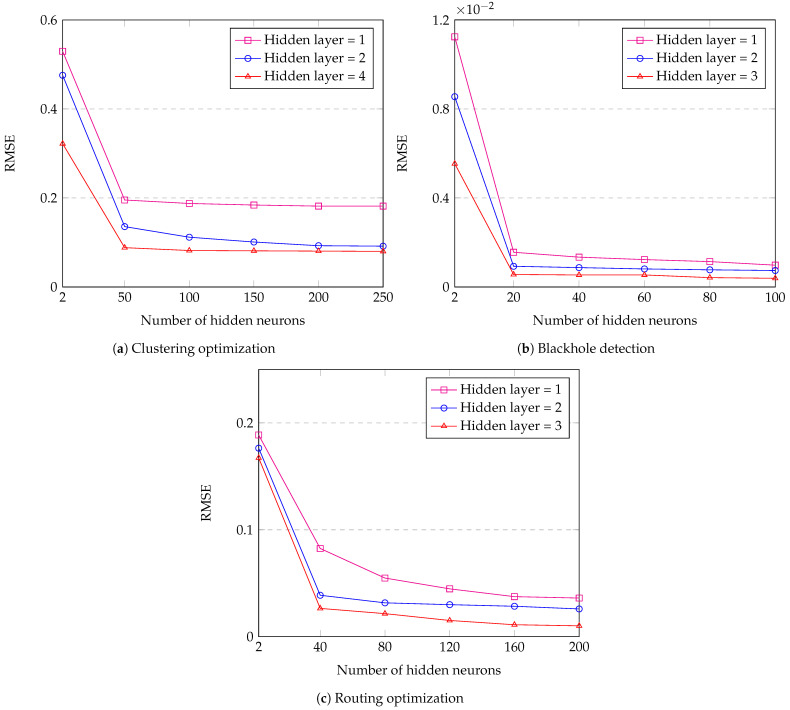
RMSE of the DL framework with different numbers of hidden layers and hidden neurons.

**Figure 12 sensors-23-08224-f012:**
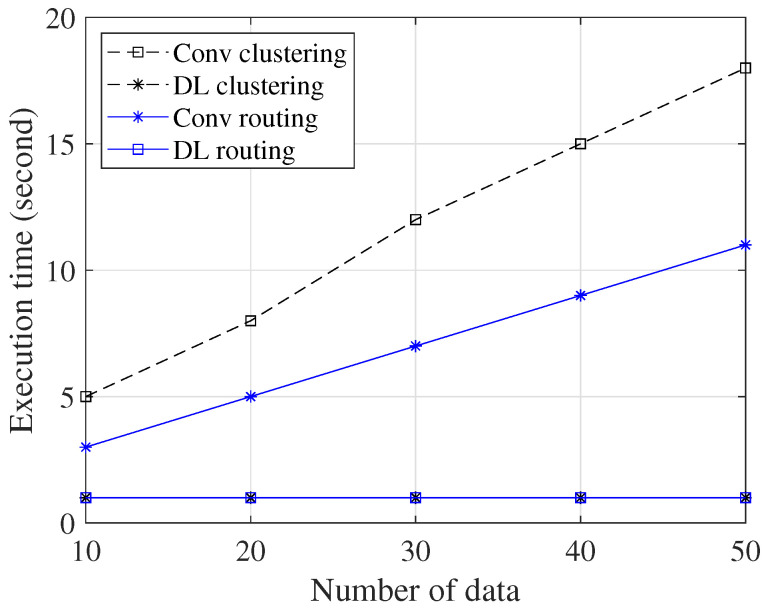
Execution time vs. variations in number of data for the clustering and routing scheme.

**Figure 13 sensors-23-08224-f013:**
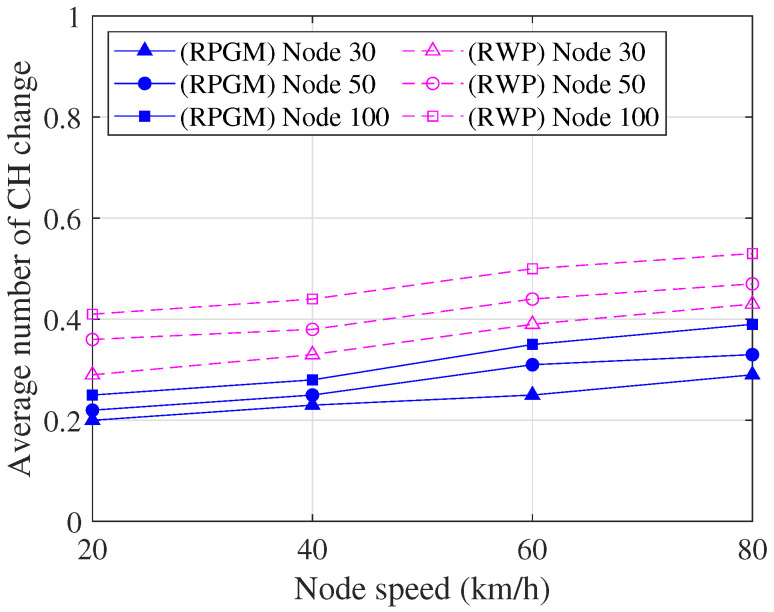
Average number of the cluster head changes as a function of node speed.

**Figure 14 sensors-23-08224-f014:**
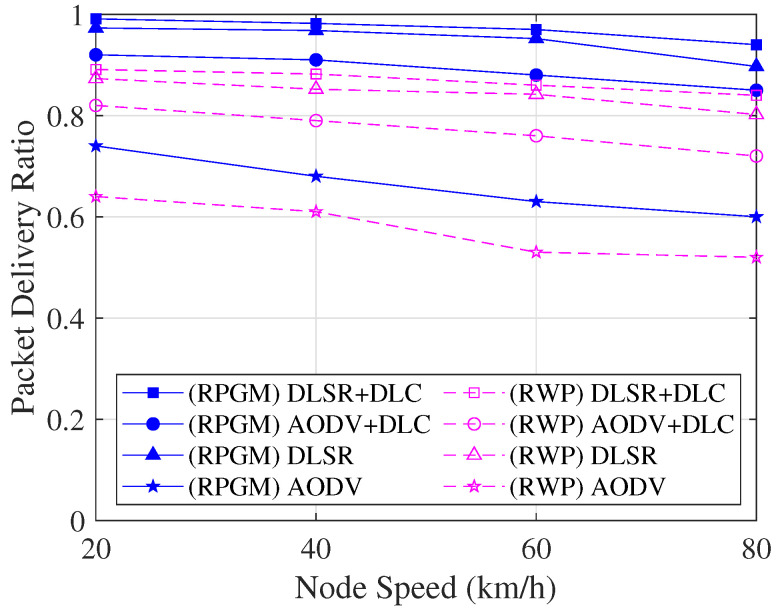
PDR with various scenarios as a function of node speed.

**Figure 15 sensors-23-08224-f015:**
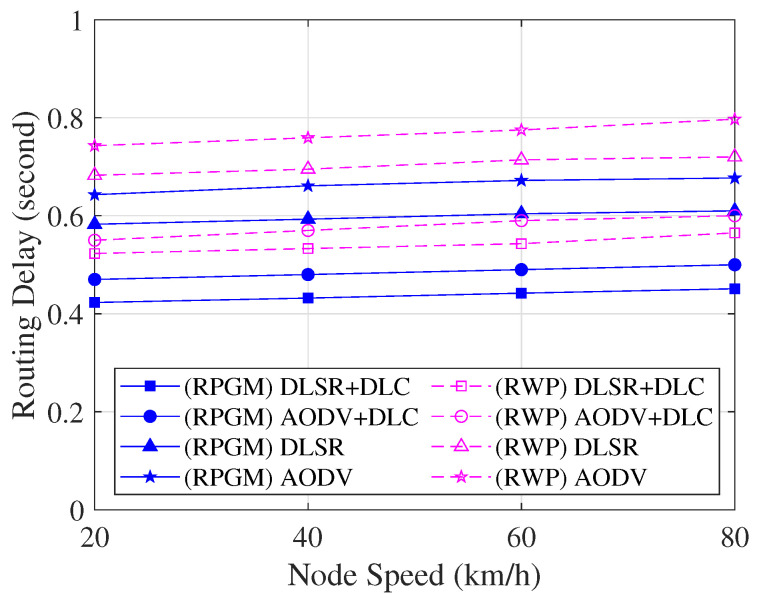
Routing delay with various scenarios as a function of node speed.

**Figure 16 sensors-23-08224-f016:**
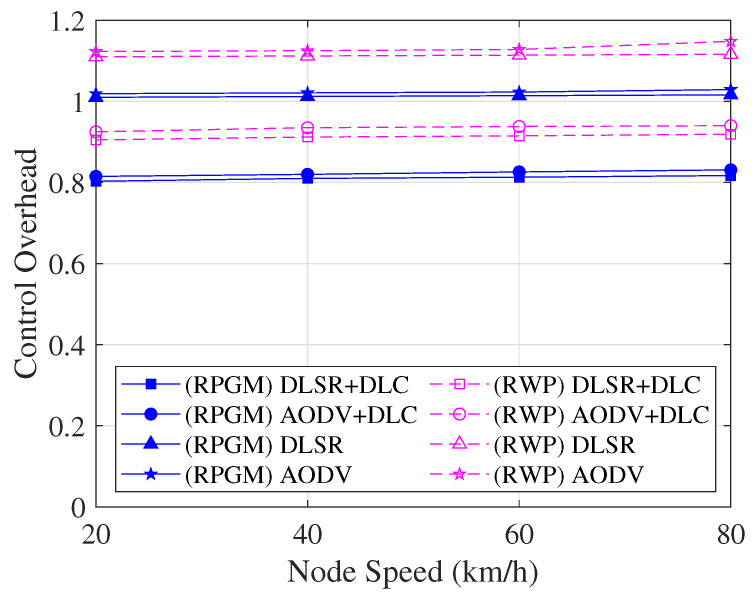
Control overhead with various scenarios as a function of node speed.

**Figure 17 sensors-23-08224-f017:**
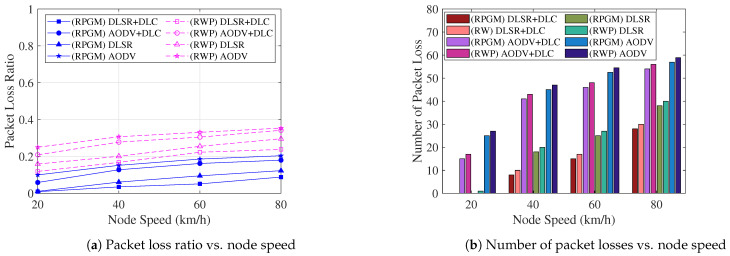
The comparison of packet loss ratio and number of packet losses as a function of node speed is performed under different scenarios.

**Figure 18 sensors-23-08224-f018:**
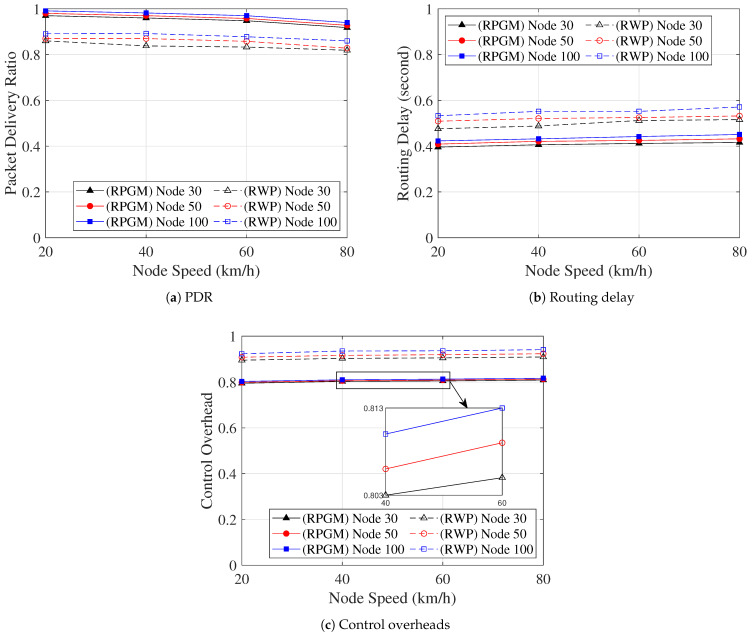
The network performance metric vs. node speed with various numbers of nodes.

**Table 1 sensors-23-08224-t001:** Summarized literature review.

Authors	Clustering	Routing	Attack Detection	Remarks
Heinzelman et al. [18]	LEACH	-	-	The cluster head is randomly chosen, regardless of their energy levels, affecting network longevity
Tselikis et al. [19]	Highest-degree-based clustering	-	Consistent clustering algorithm to classify suspicious nodes	Does not consider other important metrics such as energy levels, which can impact the network’s overall performance
Nguyen et al. [20]	Lowest-ID Algorithm	-	-	Does not take into account any metric other than the ID itself
Bao et al. [21]	PSO	PSO	-	The iteration-based algorithm spends so much time finding the optimal cost
Giri et al. [22]	Optimized Fuzzy Clustering Algorithm	PSO	-	The iteration-based algorithm spends so much time finding the optimal cost
Sahoo et al. [23]	Genetic Algorithm	-	-	The iteration-based algorithm spends so much time finding the optimal cost
Shashwat et al. [24]	-	mAODV	Intrusion detection system for blackhole attack	The false-positive detection in a harsh environment may occur
Kadam et al. [25]	-	D&PMV	Distrust value for malicious node detection and prevention.	Requires more time for processing, resulting in high end-to-end delay
Purohit et al. [12]	-	AODV, ZRP	Encrypted random number for blackhole attack detection	Additional fields in the control packets for cryptography algorithms lead to substantial routing overhead and increased end-to-end delay
Polat et al. [26]	-	-	SSAE and softmax classifier deep network schemes for DDos attack detection	Failed to improve the security of SDN-based VANETs using hardware application with limited resources
Alsarhan et al. [27]	-	-	SVM using three ML algorithms (GA, ACO, and the PSO) for classifying intrusion	Failed to include large amounts of data obtained via vehicular communication for SVM training
Velayudhan et al. [28]	Modified K-harmonic means clustering	-	CMEHA-DNN for identifying Sybil attack	Unable to identify different types of attacks in VANETs
The proposed approach	DLC	DLSR	DNN for blackhole attacks detection	Using DL in clustering, routing, and blackhole attacks detection improves network security and efficiency by delivering accurate results and very different from conventional optimization methods that involve iteration processes

**Table 2 sensors-23-08224-t002:** Clustering table of the proposed DLC protocol.

$N_{\mathrm{ID}}$	Stat	${\mathrm{CH}}_{\mathrm{ID}}$	CM

**Table 3 sensors-23-08224-t003:** Packet list for the DLC protocol.

Packet Name	Stand for	Field Information
INFO	Information Packet	$\mathrm{Type},S_{\mathrm{ID}},D_{\mathrm{ID}},E$
CHI	Cluster Head Info Packet	$\mathrm{Type},S_{\mathrm{ID}},D_{\mathrm{ID}},\mathrm{Pos},S,\mathrm{Dir}$
JC	Join Cluster Packet	$\mathrm{Type},S_{\mathrm{ID}},D_{\mathrm{ID}},\mathrm{Stat}$
AC	Accept Cluster Packet	$\mathrm{Type},S_{\mathrm{ID}},D_{\mathrm{ID}}$

**Table 4 sensors-23-08224-t004:** Routing table of the proposed DLSR protocol.

PN	NX	F	$N_{\mathrm{ID}}$	$S_{\mathrm{ID}}$	$D_{\mathrm{ID}}$	$B_{\mathrm{ID}}$	$S_{\mathrm{seq}}$	$D_{\mathrm{seq}}$	H

**Table 5 sensors-23-08224-t005:** Packet list for the DLSR protocol.

Packet Name	Stand for	Field Information
RREQ	Route Request	$\mathrm{Type},S_{\mathrm{ID}},D_{\mathrm{ID}},S_{\mathrm{seq}},D_{\mathrm{seq}},B_{\mathrm{ID}},H$
RREP	Route Reply	$\mathrm{Type},D_{\mathrm{ID}},S_{\mathrm{ID}},D_{\mathrm{seq}},E,\mathrm{Pos},H,F$

**Table 6 sensors-23-08224-t006:** The layer structure of the DNN model for clustering weight maximization.

	Size	Activation Function
Input	5	-
Layer 1	100	ELU
Layer 2	150	ELU
Layer 3	200	ELU
Layer 4	100	ELU
Output	3	LINEAR

**Table 7 sensors-23-08224-t007:** The layer structure of the DNN model for node classification.

	Size	Activation Function
Input	10	-
Layer 1	40	ELU
Layer 2	100	ELU
Layer 3	80	ELU
Output	2	LINEAR

**Table 8 sensors-23-08224-t008:** The layer structure of the DNN model for routing weight minimization.

	Size	Activation Function
Input	3	-
Layer 1	120	ELU
Layer 2	200	ELU
Layer 3	80	ELU
Output	3	LINEAR

**Table 9 sensors-23-08224-t009:** Simulation environments and network parameters.

Parameters	Value
Simulator	NS3
Simulation area	1000 × 1000 $m^{2}$
Simulation time	1000 s
Packet size	1024 bits
Mobility model	RPGM and RWP
Session length	5 s
Number of nodes	[30, 50, 100]
Node’s speed range	[20:20:80] (km/h)
Transmission range	250 m
Receive signal strength indicator (RSSI) threshold	−80 dBm
MAC protocol	802.11a

**Table 10 sensors-23-08224-t010:** DNN training parameters.

Parameters	Value
Dataset	100.000
Epoch	50
Batch size	256
Optimizer	Adam
Learning rate	0.00001

## Data Availability

Not applicable.

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
