# Peer review of "A Deep-Learning-Based Secure Routing Protocol to Avoid Blackhole Attacks in VANETs†"

_sensors, 2023, doi:10.3390/s23198224_

Round 1

Reviewer 1 Report

(1) It is recommended to put some space between the picture captions and the content of the article. See figure10.

(2) By comparing the experimental situation of more protocols and methods, the superiority of DLSR and DLC in avoiding black hole attacks can be better illustrated.

(3) Please briefly describe why the performance of the proposed clustering protocol and routing should be evaluated by those specified metrics in section 5.2.

(4) This article mentioned that the increase in node speed and number of nodes can lead to higher control overhead, it is best to discuss how to solve this problem.

  • There are not many flaws in the quality of the English language in this paper

Reviewer 2 Report

Few points should be considered for the next review stages.

  1. In the abstract, contributions summarise the methodologies. Should have the specific contribution of the article, not the process.
  2. At the end of the introduction, the authors suggest DL is the best technique for the approach and use some citations. This is a very generic statement. Authors should mention the specific point to which they are contributing to this article.
  3. A literature review should have a concrete research gap that the authors are trying to solve.
  4. Section 5 should have testbed details, including parameters, fine tuning, etc., so that it can be reproduced.

Reviewer 3 Report

The authors of this manuscript present "A Deep Learning-Based Secure Routing Protocol to Avoid Blackhole Attacks in VANETs". However the following observations have made to improve the qualty of the manuscript.

1. Define the abreviation one time and use it in the text. I noticed that the the abrevations are difine again and again. For example VANETs is defined in line 1, 19 and 133. 

2. line 64, the related work is week. Compare the related works in a table and find the research gap.

3. Research question is not defined.

4. What is the reference of equation (1)?

5. Figure 6 and 7 show very basic architecture of DL. 

6. Discuss the results before conclusion. Explain, how they are useful then previous studies. 

Minor editing of English language required

Round 2

Reviewer 2 Report

Suggest to polish another few rounds. 

Reviewer 3 Report

The authors of this manuscript have revised this manuscript according to my previous comments. Now, I am agree to accept this manuscript in its present form.

Minor editing of English language is required.